# Self-correction in science: The effect of retraction on the frequency of citations

**Anton Kühberger** [1,2]*, **Daniel Streit**[1], **Thomas Scherndl**[1]

**1** Department of Psychology, University of Salzburg, Salzburg, Austria, **2** Centre for Cognitive Neurosciences, University of Salzburg, Salzburg, Austria

* anton.kuehberger@plus.ac.at

**Data Availability Statement:** All relevant data are within the paper and its Supporting information files.

## Abstract

We investigate the citation frequency of retracted scientific papers in science. For the period of five years before and after retraction, we counted the citations to papers in a sample of over 3,000 retracted, and a matched sample of another 3,000 non-retracted papers. Retraction led to a decrease in average annual citation frequency from about 5 before, to 2 citations after retraction. In contrast, for non-retracted control papers the citation counts were 4, and 5, respectively. Put differently, we found only a limited effect of retraction: retraction decreased citation frequency only by about 60%, as compared to non-retracted papers. Thus, retracted papers often live on. For effective self-correction the scientific enterprise needs to be more effective in removing retracted papers from the scientific record. We discuss recent proposals to do so.

## Introduction

Occasionally, the scientific literature needs correction. Since errors in scientific publications come in various ways, correction also can take a variety of approaches. The smoothest way is critique in its different forms, often following from non-replication. Critique does hardly change the value of the original contribution but can change its interpretation. A less smooth way of eliminating error is to publish a corrigendum (or erratum) of parts of a paper. If there is an expression of concern for a paper, something is terribly wrong with a paper. The hardest method–the nuclear option—is retraction of a whole paper. A retraction indicates that a peer-reviewed original paper is invalid as a source of knowledge: it should be completely withdrawn from the scientific record. According to the Committee on Publication Ethics (COPE; https://publicationethics.org/files/retraction%20guidelines.pdf), reasons for retraction are: (i) clear evidence for misconduct or honest error; (ii) duplicate publication without proper reference; (iii) plagiarism, and (iv) unethical research. If there is suspicion that any of these reasons applies, but cannot be proven, journals should issue an expression of concern. Finally, a paper should be corrected, if (i) a small proportion of otherwise reliable information is misleading, or (ii) the author list is incorrect. Irrespective of the proposal of COPE, the correction of the publication record comes under many different names: retraction; correction; withdrawal; removal; expression of concern; erratum; or corrigendum.

**Funding:** The author(s) received no specific funding for this work.

**Competing interests:** The authors have declared that no competing interests exist.

Retractions proper are rare and the reasons for retraction frequently remain opaque, although in general, retraction notices should contain information on the reason(s) for retracting a paper. It seems that about 60% of retractions are due to research misconduct (fabrication, falsification, or plagiarism) [1], although nearly 40% of retraction notices fail to mention fraud or misconduct. If identified, about 60% of retractions are initiated by author(s), about 20% by editors, about 5% by journals, 15% by publishers, and less than 1% by institutions [2]. If the retraction of a paper is indicated by a retraction notice, reasons for the retraction are often provided. Retraction notices exists for most of the retracted scientific articles. In a most welcome enterprise, *Retraction Watch* (https://retractionwatch.com) collects and analyzes retractions. *Retraction Watch* lists about 39,000 retraction or other notices by october, 2022. Among those are more than 260 retractions and some expressions of concern related to the Coronavirus. This is frightening, given the short time the latter topic is virulent. The proportion of retractions was increasing around the beginning of the millennium and was evaluated to be plateauing in recent years by some authors [3, 4]. The period between the early 1980s and 2009 saw a tenfold increase in the proportion of retractions from 0.02% in the early 1980s to 0.2% in 2005–2009 [5]. Similarly, van Noorden [6] reports that the proportion of retractions rose by a factor of 10 between 2001 and 2010 (from about 40 to about 400 per year), as compared to a much smaller increase in published papers (about 44%; to 1.4 million per year in 2010). Current estimations for the proportion of retractions are about 4 in 10,000 [2]. However, the proportion of retractions depends on the publication outlet [7], with more important journals seeing more retractions. For instance, the journal *Science* has a retraction rate of 0.34% in the period between 1983 and 2017 [8].

An actual count of retractions or other correction notices in *Retraction Watch* gives an overview over the development (see Fig 1). Inspection of Fig 1 shows that the number of retractions and other notices is steadily increasing over the last 20 years. Even a linear model (which is incorrect as the increase seems to be exponential) shows a correlation between retraction year and number of retractions of r = 0.69. Note also the two extreme counts in the years 2010 and 2011, where one publisher (IEEE) has retracted thousands of meeting abstracts.

Retraction is *the* correction method: "if the problem is sufficiently serious, calling into question the main findings of a piece of work, the most effective route of correction is through retraction of the paper itself" (Nature, 2003, p.1). It is unclear, however, how effective the retraction method is for correcting the literature. Indeed, some authors report that retracted papers still are cited affirmatively [9]. One paper found no substantial difference of citation counts to retracted and comparable non-retracted articles [10], another reports an increase in post-retraction citations compared to the number of citations received before the retraction [11]. Other authors, in contrast, point to the effectiveness of retractions [1, 12, 13]. Overall, however, authors lament the lack of effectiveness of retractions for decreasing citations, for instance in dentistry [14, 15], engineering [16], biomedical research [7, 17], oncology [18], or the humanities [19].

Simply counting citations is a problematic measure of effectiveness, since citation counts follow a specific temporal pattern, including a peak within the first few years, followed by a decline after five to ten years after publication [13, 20]. Since the average time from publication to retraction is between two and four years [21], a decline in citations may be incorrectly attributed to the retraction, rather than to the usual pattern seen in the declining phase. To evaluate the effect of retractions, a control group of non-retracted papers undergoing the same citation pattern is needed. Some authors [e.g., 9, 17, 22–24] did such studies. Their control sample consisted of papers published immediately before and after the retracted paper. These studies have small samples or are outdated now. Thus, sceptics remain skeptical, mostly

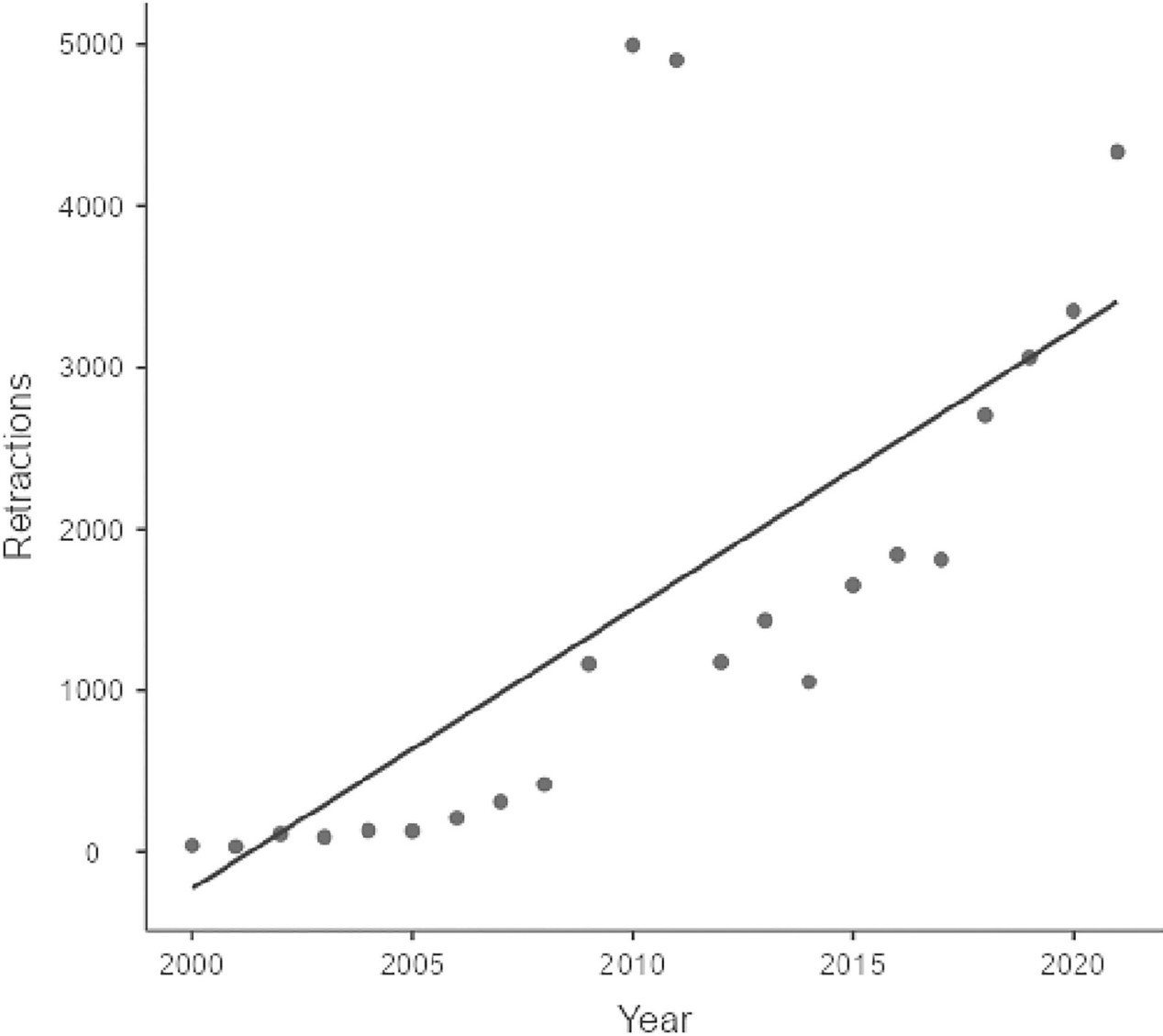

**Fig 1. Number of retractions and other notices from 2000 to 2021 according to *Retraction Watch*.** Linear trend superimposed.

because of the recent findings of continued affirmative citation after retraction [7, 24, 25]. Decisive evidence is necessary. The current study provides such evidence.

## Materials and methods

The basic design of our study consists in: (i) identifying retracted papers and their corresponding retraction notices; (ii) selecting a matched control group of non-retracted papers; (iii) counting the citations to both groups for maximally 10 years, centered at the year of retraction retrieved from the retraction notices; and (iv) comparing citation frequencies before and after retraction in both groups. We ran a pilot study (without the control group of non-retracted papers) and did a first attempt of combining databases. This served for finding out the best way to achieve our goal. Finally, we used the *PubMed* search engine to identify retracted articles and to collect their bibliographic data. By may 4, 2018, the *PubMed* web-service was

queried for "Retracted Publication" to download info on retractions, and by "Retraction of Publication" to download info on retraction notices. This resulted in 5,876 retracted articles, and 6,069 retraction notices. We combined these samples, cleaned for duplicates, and ended up with 5,663 papers identified as retracted by may 2018. Using *Scopus* we obtained a detailed count of annual citations to 4,500 of the retracted papers. After cross-checking on correct identification, we ended up with 4,159 retracted papers for which we obtained bibliographic information, the year of retraction, and citation counts for maximally five years prior and five years after the retraction.

Next, we selected control papers. For each retracted paper a matched paper was selected by randomly sampling from the same journal and same year. This was done in *Scopus*, and resulted in 3,383 articles, since in many cases the automated search did not return a hit due to various reasons. Cross-checking for correct identification and deleting duplicates resulted in 3,240 papers for which we obtained bibliographic data and annual citation counts.

## Results and discussion

The mean number of citations to retracted articles overall (before and after retraction, excluding articles that were never cited) was 36.5. The median number of citations was 15, with a minimum of 0, and a maximum count of 1130 citations. Mean time to retraction was 3.9 years, median time was 2 years, with a range from 0 to 26 years to retraction. Fig 2 depicts the frequency of retractions per year, and Fig 3 shows the time lag between publication and retraction. Most papers were retracted soon after publication, with about ¼ being retracted in the

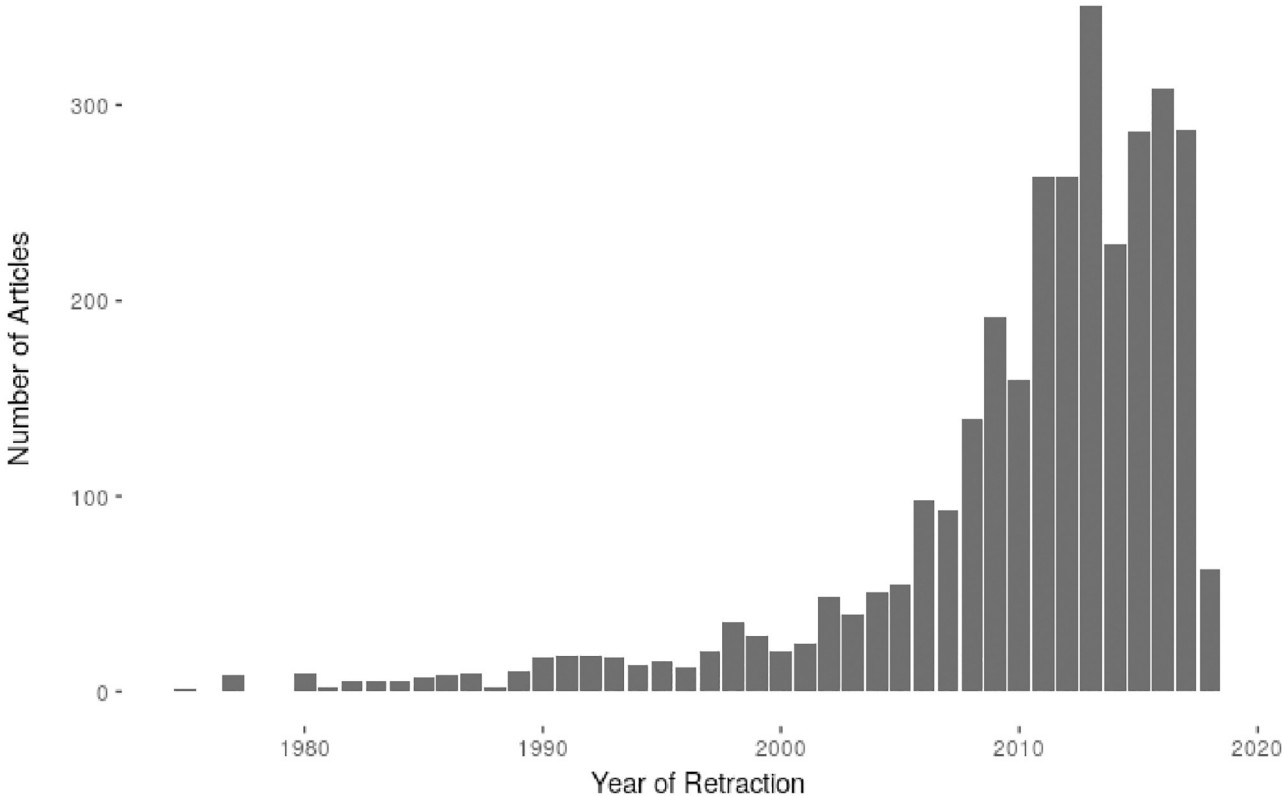

**Fig 2. Frequency of retractions by retraction year.**

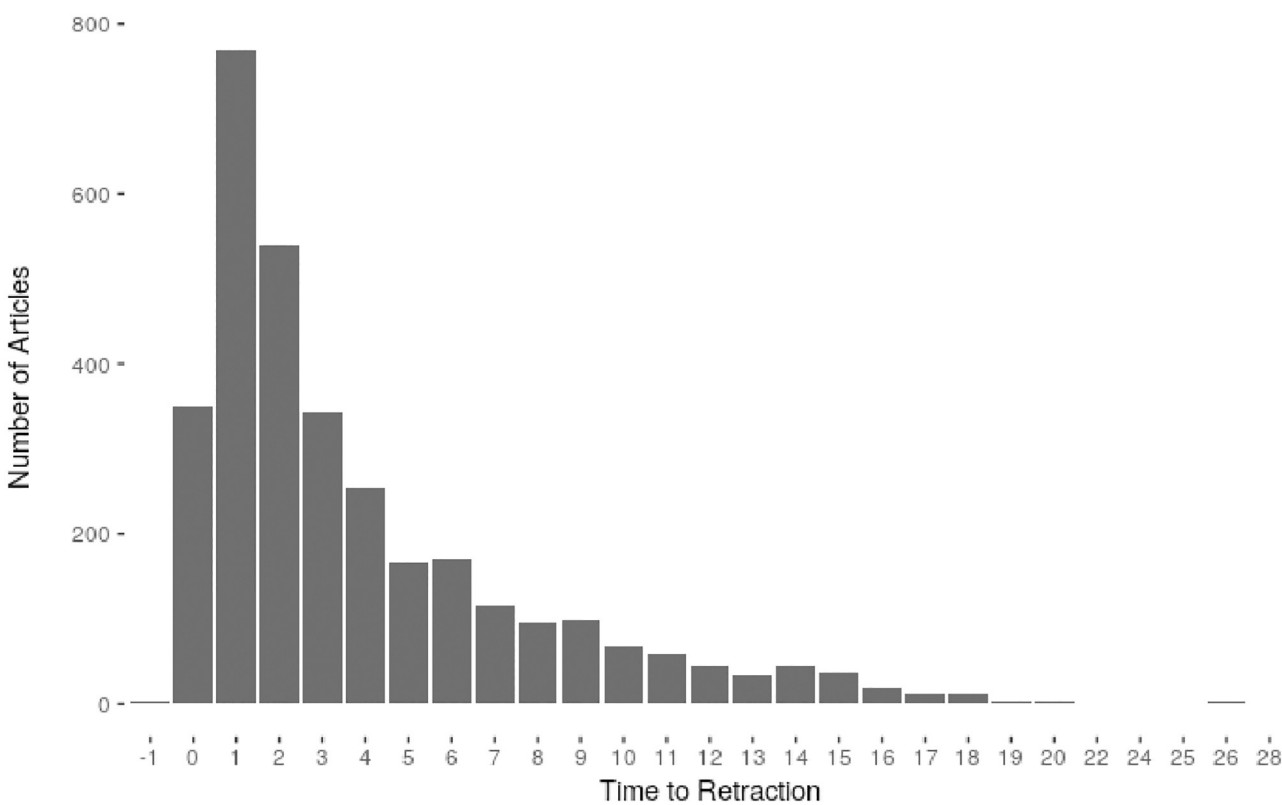

**Fig 3. Time lag between publication and retraction.**

year following publication. Note, however, the large variability in time lag between publication and retraction.

To measure the effect of retraction on citations, we ran a series of hierarchical linear mixed-effects models using the *lme* function of the *nlme* R-package [26]. Mixed effects models have the advantage of allowing a good deal and varying patterns of missing data, and of modelling correlated data [27]. Both these advantages are important for our data, where citation frequencies of papers are correlated, and the length of the pre-retraction period varies for papers between 1 and 5 years, leading to missing data. Stepwise, we added 'retraction' (paper retracted vs not retracted); 'period' (paper cited before vs after retraction), and the interaction of 'retraction' and 'period'. The interaction directly investigates the main research question, since only in the period after retraction, retracted papers ought to be cited less frequently than non-retracted papers. Table 1 gives the results of the hierarchical steps. Most notable is the last step, showing that adding the interaction improved the model with the significant main effects substantially ($\Delta\chi^2(1) = 793.60$, p < .001).

In terms of simple counts, retracted papers had a mean yearly citation count of 4.85 before retraction, decreasing to 1.93 after retraction. In contrast, non-retracted papers had, in the same period, a mean yearly citation count of 3.88, which increased to 4.73. Recall that the periods (pre vs post) of retracted and control papers are matched, enabling a direct conclusion about the effect of a retraction: a retraction led to an estimated average loss of about 2.8 (4.73–1.93) citations annually. Indeed, the effect might even be stronger, since retracted papers scored about one citation higher in the period before retraction than did control papers (see

**Table 1. Results for mixed linear effects models.**

| Model | df | AIC[a] | BIC[b] | logLik[c] | Test | L.Ratio[d] | p-value |
|---|---|---|---|---|---|---|---|
| 1) Baseline | 4 | 300412 | 300447 | -150202.1 | | | |
| 2) 1 + Retraction | 5 | 300045 | 300088 | -150017.5 | 1 vs 2 | 369.2 | < .001 |
| 3) 2 + Period | 6 | 299849 | 299901 | -149918.4 | 2 vs 3 | 198.2 | < .001 |
| 4) 3 + Retraction x Period | 7 | 299057 | 299118 | -149521.6 | 3 vs 4 | 793.6 | < .001 |

Note.

[a] Akaike-Information-Criterion.

[b] Bayesian-Information-Criterion.

[c] log-likelihood;

[d] Likelihood Ratio.

Fig 4). Retracted papers also lost their initial advantage; adding this leads to an estimated loss of about four citations each year, compared to non-retracted papers.

We also calculated a more complex model, by replacing the general pre/post predictor by the five single years before, and five single years following retraction (see Fig 5). The pattern of findings also held for single years: (i) in the pre-retraction period retracted papers had a higher citation count (by about 1 citation each year) than matched non-retracted papers; and (ii) in the post-retraction period retracted papers had a lower citation count (by about 3 citations each year) than matched non-retracted papers. Fig 5 seems to show a discontinuity in the retraction year. This is because citation counts in the retraction year are not depicted. The discontinuity thus indicates that a retraction decreases citations already in the retraction year. In sum, retraction led to a loss in average annual citations of about 60% (from 4.85 to 1.93), while non-retracted papers received an increase in citations of about 20% (from 3.88 to 4.73) in the same period of a paper's scientific life. Expressed in numbers: in the first 5 years after publication non-retracted papers were cited about 4 times annually. Papers that later were retracted received about 5 citations before retraction, that is, they initially had about 25% more citations. In contrast, after retraction, citations dropped to about 2, while increasing to about 5 for non-retracted papers. Note also that some of the citations to retracted papers are negative, citing the paper because it was retracted rather than as valid evidence. Limited evidence indicates that the frequency of negative citations seems to be rather low, being well below 10% [7, 10, 24].

We ran a subgroup analysis to gain more detailed information on how citation patterns develop over time. Specifically, we compared papers that were retracted more or less immediately (within less than 2 years after publication) to papers that remained longer in the market before being retracted (later than 2 years since publication). The retraction penalty is evident in Fig 6: early retracted papers (average yearly citation count = 0.62) and their control papers (average yearly citation count = 0.57) did not differ in citations immediately after publication (t(21825) = -0.14, p = .99), that is, they had the same start. Early retraction showed a strong effect on subsequent citations, however (average yearly citation count = 1.53, and 5.09, for retracted and control papers, respectively; t(21825) = 25.72, p < .0001). That is, as expected, citations strongly increased for control papers, but hardly increased for retracted papers. For late retractions the picture was different. Both groups had attracted many pre-retraction citations, with the retracted papers being in the lead (5.46 vs. 4.40 annual citations; t(21825) = -9.92, p < .0001). This changed after retraction, where the retracted papers underwent a considerable loss in citations (2.15 citations), compared to non-retracted controls (4.50 citations; t(21825) = 22.15, p < .0001).

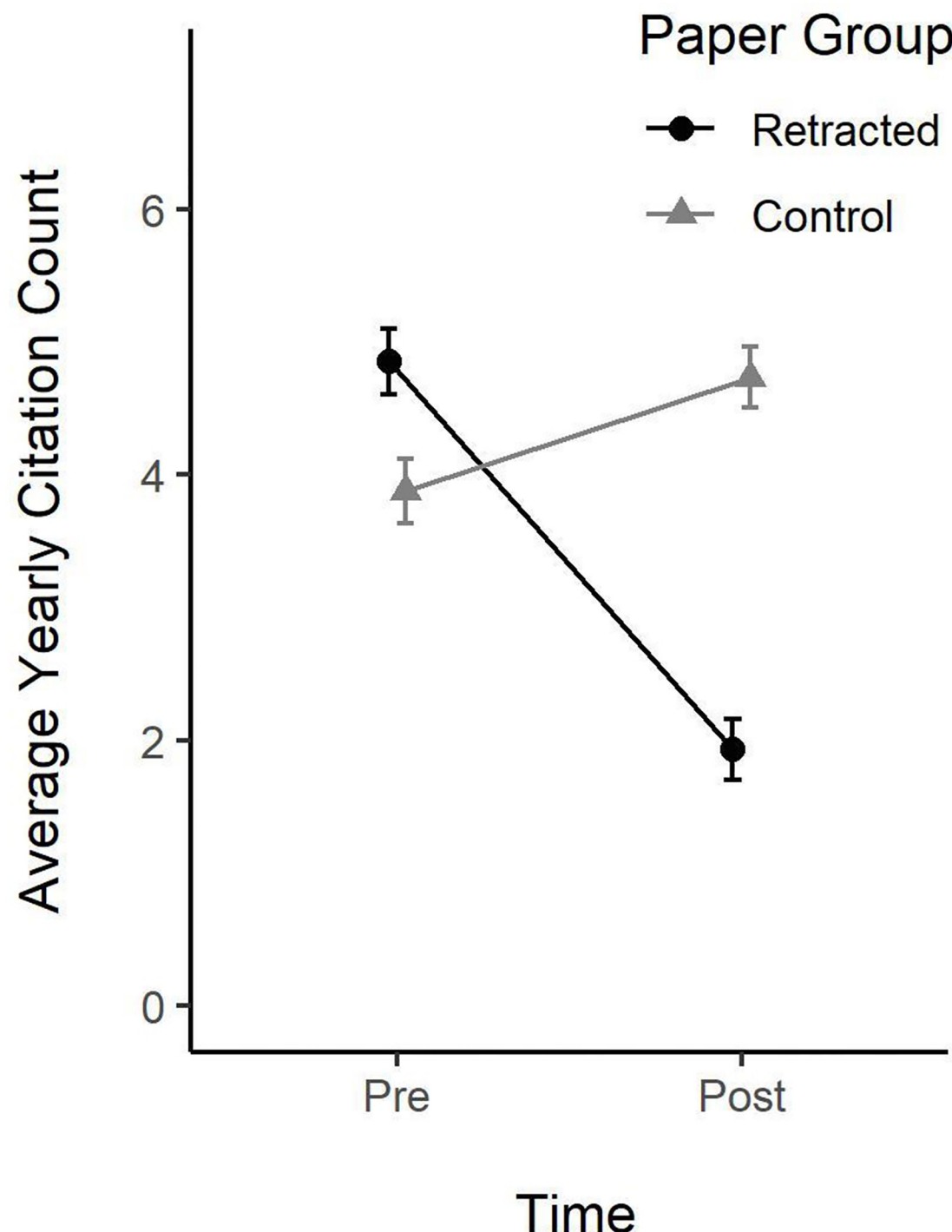

**Fig 4. Retraction by period of publication.**

## Discussion

Our study adds to the evidence on the limited effect of retractions on citations: retractions decrease citations, but do not eliminate them [4, 7, 14, 18, 22, 25, 28]. Here we found an

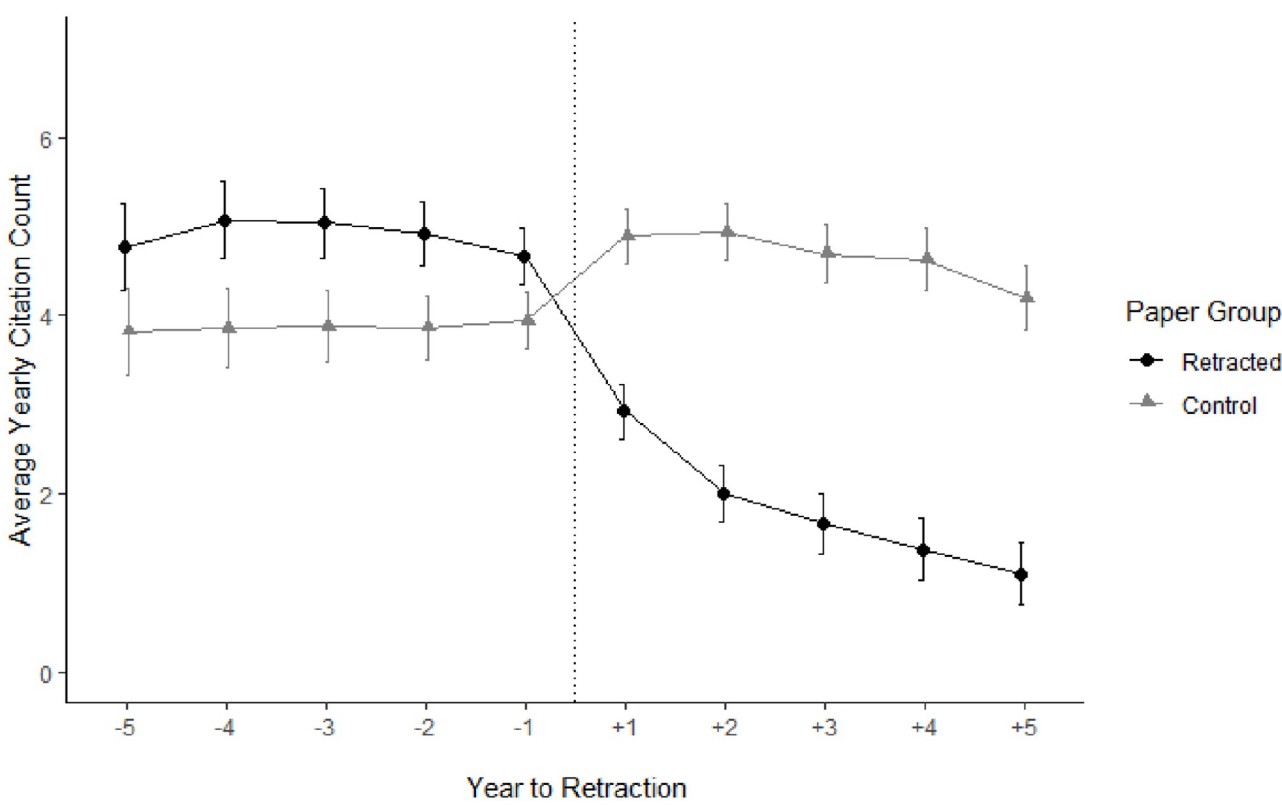

**Fig 5. Retraction by time of publication, for single years.**

average annual citation count of about 2 for retracted papers, indicating that retracted papers still live on. However, we also found a strong and direct loss of citations due to retraction, cutting citations down to less than half during the following years. These findings are in line with others, also reporting retraction losses [4, 22, 24, 28], and the size of the loss is comparable over studies. Distinguishing between papers that are retracted shortly after publication and papers that are retracted more than two years after publication, shows that the retraction penalty exists in both phases of a scientific paper. In the early phase retraction alleviates citation, such that papers do not live up to their potential. In the late phase retraction decreases citation, such that retracted papers experience considerable citation losses, compared to controls in the publication market. Taken together, this testifies to the strong, although not absolute, effect of retraction on citations. Interestingly, the effect for failure to replicate on citations seems to be much weaker, if at all existing [29–31]. As a parallel to our findings, non-replicating papers are cited more than those that are replicable: we also found that retracted papers were cited more than non-retracted ones before retraction, unless they were retracted immediately after publication.

Obviously, retracted papers carry a negative stigma. However, there is something special to these papers, because they are cited more frequently than others before retraction, or even generally [32], provided they have been in the market for some years. We do not think that the 25% advantage in citations found here is due to some uninteresting artifact. Rather, the reason is that retraction of a paper presupposes it being perceived and probably read, to begin with. Only the important (i.e., well cited) papers have some potential to become the target of scientific investigation, be it for replication, or for methodological and statistical criticism. In an

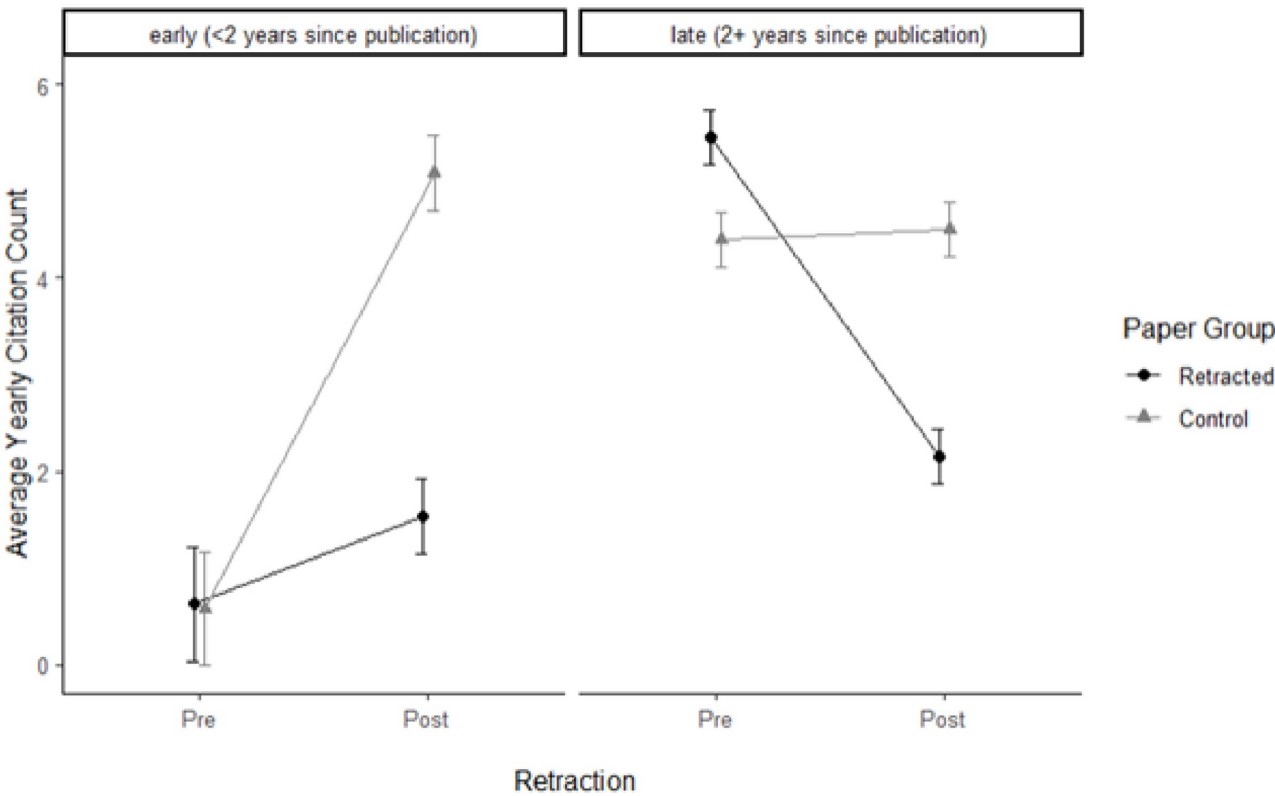

**Fig 6. Retraction by time of publication, for papers immediately (early), and later retracted (late).**

ironical twist, one could say that to qualify for retraction, papers ought to be good, or at least special, in some sense. It is also possible that the initial advantage in citations is related to the author's reputation, or, even more likely, to the number of authors [33]. The speculative reasoning goes like this: Cooperation enables the acquisition of bigger projects and more resources. These projects generate more pressure to come up with positive results, and, lacking adequate quality control, they do so, even if questionable research practices are necessary to ensure publication.

We think that our study has high validity and broad implications. First, our sample contains several thousands of papers, with their citation history measured over a time interval of 10 years, delivering a massive amount of data. Second, by including a noticeable percentage of retracted papers, the sample–to–population relationship is unprecedented. Third, our study is easily interpreted, since we report the unstandardized effect size of a retraction: on average, a retraction results in a yearly citation loss of about 3 citations. In relative terms: a retraction leads to a loss of more than 60% citations. Retraction immediately after publication has an even stronger effect on citations. Thus, even if retractions often do not have negative legal or financial consequences, their scientific penalty is high. Indeed, the penalty does not only apply to the retracted paper, but also to other papers of the same authors, and even to the papers they have written before the incriminated one [32].

Qualitatively, the retraction penalty results in a decreasing pattern of citations, while otherwise the pattern of citations would still be increasing, at least for some years. This is what our findings show, and the temporal citation pattern for the control group is as expected from available research. For instance, Johnston et al. [20] investigated the economics literature, and

found an increasing frequency of citations until about year fife after publication, followed by a decrease in later years. For empirical papers, the citation counts in [20] are comparable to our findings, with a peak of about 5 citations annually around year fife. Theoretical papers earned less citations. As a side aspect: theoretical papers presumably have only a small chance of being retracted, isn't it?

The effect of retraction on citations can vary widely. For instance, one of the top 10 retracted papers [34], according to *Retraction Watch* (accessed oct 2022), is an example for the lack of effectiveness of retraction. It was retracted in 2007, two years after its publication. Before retraction, it was cited 232 times, after retraction it was cited 1192 times, very often not even indicating that the original paper was retracted. Consulting *Scite_* (https://scite.ai/), a website that enables the evaluation of scientific articles by describing whether a citation is supporting, contrasting, or simply mentioning a paper, underscores another finding with respect to retractions: many more citations are supporting than contrasting, even for retracted papers. In the case of the Fukuhara et al paper [34], *Scite_* reports the following frequencies (oct 2022): 1021 mentioning citations, 28 supporting, but only 8 contrasting citations (51 citations unclassified). Surprisingly, when accessed via the journal *Science*, this paper still is not marked 'retracted'; only when downloading the pdf of the paper the retraction is indicated and a retraction notice is appended to the pdf. The *PubMed* entry of the paper clearly indicates its retraction. As another example, take the well-known Wakefield study on MMR-vaccination [35], which is #2 on the *Retraction watch* hitlist. It was published in 1998 and fully retracted in 2010. In the 12 years before retraction it collected 642 citations, in the 12 years after retraction it collected 867 citations. Only 8 of all citations were contrasting, according to *Scite_*. These examples testify to the limited effect of retraction on citations. Note, however, that little is known about the reasons why some retractions are effective in stopping citations while others fail. Our analysis only conveys the average effect of retractions on citations and is mute about the effects variability and source.

Retraction of a paper can follow from a variety of reasons, ranging from fraud to publisher error. Some reasons for retraction do not automatically invalidate the evidence reported in a paper. A first action to decrease the damage done by faulty research is to point out whether the retraction indeed indicates that the findings are wrong. For instance, if a paper is retracted because the order of author is incorrect, this does not render the paper worthless. The label retraction, with its negative connotation, should be saved for real misconduct, and should not encompass inadvertent error. This would ease self-correction for the latter, which is laudatory, and should not carry a negative stigma [36].

Many options for improvement in science publishing center around either correcting scientists' minds or changing publication practices. The former include recommendations for correct reporting [37], or training scientists on responsible conduct of research [38]. In addition, they include increasing awareness of the retraction status of papers, for instance by reliance on reference management software [39]. Some free referencing tools signal the retraction status of papers (e.g., *Zotero*, *Pubpeer*). Options for changing publication practices around retractions include prompt retraction, detailed retraction notices, and establishing monitoring and alert systems to track retraction [40], or optimizing peer review [41, 42]. Some ideas are still broader, for instance by encouraging the publication of negative results and thereby reducing pressure to publish [43] or changing the incentive structures of researchers [44].

A most critical point for avoiding citations of retracted papers is that those papers should never be cited, irrespective of the time of retraction. If a paper is untrustworthy, this applies as soon as it is published. It should not be published at all. However, only starting from the date of retraction we know about this and can avoid citation. Thus, the problem runs much deeper than the about 150.000 citations to papers after retraction, that were estimated by Dinh et al.

[45] in 2019. The literature needs to be purged from all citations to retracted papers, as soon as a retraction becomes public. However, we can conceive of no method to correct the literature backwards, getting rid of citations that date before the retraction. Ultimately, what can help here is only to promote an open research culture, valuing transparency, openness, and reproducibility [46]. Thus, Fanelli [47] makes a good point when he argues for redefining misconduct as distorted reporting, rather than forged acting in the form of fabrication, falsification, and plagiarism. In essence, trust in individual researchers must be replaced by a model of science where the system is inherently trustworthy [48].

## Conclusion

Retraction is the strongest form of correction in science. To fulfill its function, the scientific community should be clearly notified of a retraction such that retracted papers should at least not be cited after retraction. In general, however, they should never be cited, which constitutes a problem inherent in the traditional system of science. We show that retraction falls even short of the goal of correcting the literature from retracted papers after retraction, since retraction fails in optimally decreasing citations. Retracted papers often live on. For substantial self-correction the scientific enterprise needs to be more effective in removing retracted papers from the scientific record. To avoid retractions altogether, the whole system for producing science, rather than only individual scientists, must become more trustworthy.

## Supporting information

**S1 Data.**
(CSV)

## Author Contributions

**Conceptualization:** Anton Kühberger, Daniel Streit, Thomas Scherndl.

**Data curation:** Anton Kühberger, Daniel Streit, Thomas Scherndl.

**Formal analysis:** Daniel Streit, Thomas Scherndl.

**Investigation:** Daniel Streit.

**Methodology:** Anton Kühberger, Daniel Streit, Thomas Scherndl.

**Project administration:** Anton Kühberger, Daniel Streit, Thomas Scherndl.

**Resources:** Anton Kühberger, Thomas Scherndl.

**Software:** Daniel Streit, Thomas Scherndl.

**Supervision:** Anton Kühberger, Thomas Scherndl.

**Validation:** Anton Kühberger, Daniel Streit.

**Visualization:** Daniel Streit.

**Writing – original draft:** Anton Kühberger, Daniel Streit.

**Writing – review & editing:** Anton Kühberger, Thomas Scherndl.

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
