## [Decision Letter · Decision Letter 0]

27 Jul 2022

PONE-D-22-16961Self-correction in science: The effect of retraction on the frequency of citationsPLOS ONE

Dear Dr. Kühberger,

Thank you for submitting your manuscript to PLOS ONE. After careful consideration, we feel that it has merit but does not fully meet PLOS ONE’s publication criteria as it currently stands. Therefore, we invite you to submit a revised version of the manuscript that addresses the points raised during the review process.

ACADEMIC EDITOR: Please go through the comments made by the reviewers. The reviewers have suggested minor edits to be made. Once these are carried out, the manuscript could be considered for publication.

We look forward to receiving your revised manuscript.

Kind regards,

Venkatesh Shankar Madhugiri

Academic Editor

PLOS ONE

Journal Requirements:

Additional Editor Comments:

Please go through the comments made by the reviewers. The reviewers have suggested minor edits to be made. Once these are carried out, the manuscript could be considered for publication.

Reviewers' comments:

Reviewer's Responses to Questions

**Comments to the Author**

1. Is the manuscript technically sound, and do the data support the conclusions?

Reviewer #1: Yes

Reviewer #2: Yes

2. Has the statistical analysis been performed appropriately and rigorously? 

Reviewer #1: Yes

Reviewer #2: Yes

3. Have the authors made all data underlying the findings in their manuscript fully available?

Reviewer #1: Yes

Reviewer #2: Yes

4. Is the manuscript presented in an intelligible fashion and written in standard English?

Reviewer #1: Yes

Reviewer #2: Yes

5. Review Comments to the Author

Reviewer #1: In the manuscript entitled "Self-correction in science: The effect of retraction on the frequency of citations", the authors describe how retraction of papers affect their citation counts. The authors obtained and analyzed the citation data over 10 years, centered around the year of retraction, for both the retracted papers and their year matched control paper counterparts from the same respective journals. The authors found that retraction caused the papers to lose on average, 60% of their citations post retraction. This was in contrast to the non retracted papers which on average, gained 20% citations in the same period. The authors noted that the retracted papers despite having a higher average citation count than the control group before retraction, lost this advantage and ended up falling lower than the control group. The effect of retraction was consistent even when yearly citation trends were analyzed. The authors conclude that even though retractions cause a significant reduction in the citation counts, they do not completely eliminate citations for the retracted papers.

The merits of the study are its original findings, robust sample size, and easy readability. The topic has broad implications, especially in the post pandemic publication landscape which has brought phenomenon of retractions, a once largely academic issue, into the view of the public. The findings of this paper are consistent with existing literature. That said, there are some revisions that can be made to the paper to make it well rounded for publication.

1) The authors report that there is a strong (but not absolute) effect of retractions on citations. And also that "retraction of a paper presupposes it being perceived and probably read, to begin with. Only the important (i.e., well cited) papers have some potential to become the target of scientific investigation". Although this sounds logical, it is still speculative. Borrowing from Hill's criteria of causality, the paper would benefit from establishing a few more measures of association that could address these speculative comments. The paper does a good job of demonstrating temporality but not specificity or consistency of association. This could be done quite easily with the data the authors have. For instance, analyzing the papers which got retracted, say <=2 years vs >2 years from publication year (since median time to retraction is 2 years), could tell us if these sets of retracted papers differ in their degree of "specialness" with respect to being read/cited. Another subgroup analysis could be to look at the retracted papers before and after 2010 (since a big spike seems to appear around that time) to see if the loss of citations is consistent. These are only suggestions and it is up to the authors to perform the necessary subgroup analyses.

2) The paper shows an increase in the control group annual citations after the retraction year (figure 4). The authors have not explained why this should be expected. Is this the natural trend of citations which increase and then decrease to a plateau? If that is the case, why is there a bump specifically after the year of retraction?

3) The authors presumably analyzed the yearly citation trends of all the papers after the retraction year for both the retracted group and the control group. This could artificially magnify the effect being studied since every year before the retraction year would not have had the same number of papers as every year after. If this was not the case, it is not very clear from the paper. The authors should report the data on the number of papers they had analyzed each year before and after the retraction year. A simple table with this information along with the mean and SD data for each year (before and after retraction) will greatly improve the clarity of the paper.

4) The paper has minor grammatical errors that do not impact readability but nevertheless could be corrected.

Reviewer #2: Authoring high quality scientific articles is an integral endeavor of scientific research. On certain occasions though, for well documented reasons, some of these scientific articles are retracted. The authors of this manuscript provide compelling data driven evidence for continued citations of retracted papers within the scientific literature. Based on their observations, clearly there is a need for the scientific journals and research scientists to take note of citations of retracted papers and create a process to identify and avoid them in their own work.

Please find below a few questions/ suggestions that could further improve this manuscript:

1. Can actionable suggestions/ recommendations be provided for ensuring that retracted articles are not further highlighted via citations. Specifically, defining the role technology could play to combat the bad practices followed by certain subset of research investigators would be appreciated by the scientific community.

2. It could be interesting to analyze the observed results as a function of scientific domains. Do certain domains tend to propagate retracted articles for longer periods of time within scientific publications?

3. Provide appropriate and adequate references to methods mentioned such as hierarchical linear mixed-effects models. Not everyone is aware of these methods and it would help interpret the results better. Especially talk about the need to use this method since they capture fixed as well as random effects.

4. Inherent biases exist in retracted publications. Some of them get wide publicized due to their relevance to current topic of interest. This was briefly touched upon by the authors. Selection biases such as this do affect the observed results. A more detailed analysis or a plan to address them would benefit the reader.

5. A larger question that arises is what can be done to ensure that high quality standards are encouraged so that the percentage of retracted papers reduce with time? As a community we need to address this so that the citations of retracted papers would be restricted.

6. Last but not the least, it is not enough to point out that there is a crucial problem plaguing the scientific publications. There is a need to strongly identify the process to capture the retracted papers during the review process and bring it forward to the journal editor's attention before accepting the same for publication.

6. PLOS authors have the option to publish the peer review history of their article (what does this mean?). If published, this will include your full peer review and any attached files.

Reviewer #1: No

Reviewer #2: No

---

## [Author Response · Author response to Decision Letter 0]

20 Oct 2022

Response to Reviewers

Title: Self-correction in science: The effect of retraction on the frequency of citations

by Anton Kühberger, Daniel Streit, & Thomas Scherndl

This letter contains our reaction to the comments of the reviewers

The questions #1 to #4 by PLoS ONE were answered affirmatively by both reviewers. We therefore respond specifically to their reactions detailed in question #5 (Review Comments).

Reviewer #1 suggested the following points to round up the paper:

1) Causality is established only by temporality, but not by specificity or consistency. The reviewer opted for establishing a few more measures of association and suggested a subgroup analysis of the retracted papers (retracted within <=2 years vs < 2 years; pp. 12-13). The revised version of the paper entails such a subgroup analysis. We found an interesting pattern of results, depicted in the new Figure 5, and discussed on pp. 12-13. The analysis offers some interesting new insights and strengthens the causality assumption. 

2) The reviewer argues that we have not explained why an increase in the control group annual citations (figure 4) should be expected. On p. 14 we do this now. In addition, we explain why there is a bump specifically after the year of retraction.

3) The reviewer argues that we should report some additional data on the papers. We do this on p. 7. Frequencies of papers and publication years are visible in Fig. 2.

4) We corrected minor grammatical errors.

Reviewer #2 suggested the following:

1) The reviewer asked for providing actionable suggestions/ recommendations for ensuring that retracted articles are not further highlighted via citations. Specifically, (s)he opted for defining the role technology to combat bad practices. We included a new section detailing what and how we think the problem could be alleviated. Broadening the reviewers comment, we include also a discussion of why not only the citation of retracted papers after retraction, but citation even before retraction is a problem, when later retraction is not yet known. We also broaden the discussion by relating the discussion to citations of nonreplicating research. The Discussion was extended by more than two pages and contains some 15 new references. 

 2) The reviewer suggests analysis scientific domains. We cannot do this since we did not systematically collect data on domains. 

3) The reviewer asks for provision of appropriate and adequate references to hierarchical linear mixed-effects models, and why we used it here. We do this on p. 9.

4) The reviewer recommends being more explicit on inherent biases in retracted publications, especially with respect to current topics of interest. We do this in the revision, in the extended discussion section (see pp. 15-17).

5) The reviewer asks what can be done to ensure that high quality standards are encouraged so that the percentage of retracted papers reduce with time? We are more explicit on this in our new discussion section (pp. 12 – 16).

6) The reviewer sees strong need to capture the retracted papers during the review process and bring it forward to the journal editor's attention before accepting the same for publication. Again, we are more specific on this in the Discussion.

In sum, we responded to the comments of the reviewers and we added some additional data and analysis. We hope the revision will be successful in making the paper publishable.

Yours sincerely

Anton Kühberger

---

## [Editor Report · Decision Letter 1]

4 Nov 2022

Self-correction in science: The effect of retraction on the frequency of citations

PONE-D-22-16961R1

Dear Dr. Kühberger,

We’re pleased to inform you that your manuscript has been judged scientifically suitable for publication and will be formally accepted for publication once it meets all outstanding technical requirements.

Kind regards,

Venkatesh Shankar Madhugiri

Academic Editor

PLOS ONE

Additional Editor Comments (optional):

The authors have addressed the comments made by the reviewers. The paper is now suitable for publication.
---

## [Editor Report · Acceptance letter]

8 Nov 2022

PONE-D-22-16961R1 

Self-correction in science: The effect of retraction on the frequency of citations 

Dear Dr. Kühberger:

I'm pleased to inform you that your manuscript has been deemed suitable for publication in PLOS ONE. Congratulations! Your manuscript is now with our production department. 

Kind regards, 

on behalf of

Dr. Venkatesh Shankar Madhugiri 

Academic Editor

PLOS ONE